Detection of antibiotic-resistant bacteria endowed with antimicrobial activity from a freshwater lake and their phylogenetic affiliation

Zothanpuia 1
Passari Ajit K. 1
Gupta Vijai K. 2
Singh Bhim P. bhimpratap@gmail.com 1
1 Department of Biotechnology, Mizoram University , Aizawl , Mizoram , India
2 Molecular Glyco-biotechnology Group, University of Ireland , Galway , Ireland , UK
Tulkens Paul
Electronic publication date: 2016 Jun 9
Publication date: 2016
Volume: 4
Electronic Location ID: e2103
Received 2016 Mar 14; Accepted 2016 May 11
Copyright: ©2016 Zothanpuia et al.
Copyright year: 2016
Copyright holder: Zothanpuia et al.
License: This is an open access article distributed under the terms of the Creative Commons Attribution License, which permits unrestricted use, distribution, reproduction and adaptation in any medium and for any purpose provided that it is properly attributed. For attribution, the original author(s), title, publication source (PeerJ) and either DOI or URL of the article must be cited.
License URL: https://creativecommons.org/licenses/by/4.0/

Keywords: BOX-PCR, Antibiotic susceptibility, PKS II, ERIC-PCR, NRPS, 16S rRNA gene

Funding: Department of Science and Technology, Government of India SERB/F/2501/2013-14 Department of Biotechnology, Ministry of Science & Technology, India Funding was provided by the Department of Science and Technology (DST), Government of India, New Delhi (for funding as young scientist project to BPS (No: SERB/F/2501/2013-14)) and the Department of Biotechnology, Ministry of Science & Technology, India, for establishment of the DBT-BIF Centre and the DBT-State Biotech Hub in the Department, which has been used for the present study. The funders had no role in study design, data collection and analysis, decision to publish, or preparation of the manuscript.

==============================
Antimicrobial resistance poses a serious challenge to global public health. In this study, fifty bacterial strains were isolated from the sediments of a freshwater lake and were screened for antibiotic resistance. Out of fifty isolates, thirty-three isolates showed resistance against at least two of the selected antibiotics. Analysis of 16S rDNA sequencing revealed that the isolates belonged to ten different genera, namely Staphylococcus(n = 8), Bacillus(n = 7), Lysinibacillus(n = 4), Achromobacter(n=3), bacterium(n = 3), Methylobacterium(n = 2), Bosea(n = 2), Aneurinibacillus(n = 2), Azospirillum(n = 1), Novosphingobium(n = 1). Enterobacterial repetitive intergenic consensus (ERIC) and BOX-PCR markers were used to study the genetic relatedness among the antibiotic resistant isolates. Further, the isolates were screened for their antimicrobial activity against bacterial pathogens viz., Staphylococcus aureus(MTCC-96), Pseudomonas aeruginosa(MTCC-2453) and Escherichia coli(MTCC-739), and pathogenic fungi viz., Fusarium proliferatum (MTCC-286), Fusarium oxysporum (CABI-293942) and Fusarium oxy. ciceri (MTCC-2791). In addition, biosynthetic genes (polyketide synthase II (PKS-II) and non-ribosomal peptide synthetase (NRPS)) were detected in six and seven isolates, respectively. This is the first report for the multifunctional analysis of the bacterial isolates from a wetland with biosynthetic potential, which could serve as potential source of useful biologically active metabolites.

Introduction

Bacteria play a vital role in benthic food web, nutrient recycling and decomposition of various organic compounds in aquatic environments (Fischer, Wanner & Pusch, 2002). Sediment is a special habitat among the aquatic ecosystem and the numbers of microbes are much higher than the corresponding water bodies (Zinger et al., 2011). The bacteria isolated from such ecosystems have an ecological significance like resistance to antibiotic which would have been adopted in the due course of selection processes (Nair, Chandramohan & Loka-Bharathi, 1992; Silva & Hofer, 1995).

One major challenge with reference to human health is the spreading and appearance of antibiotic resistance among the pathogens (Martinez & Baquero, 2014; Huang et al., 2016). This can be tackled by the discovery of new antibiotics having an alternate mode of action which can eliminate disease causing pathogenic microbes. Screening of microorganisms from their natural habitat is an important step for the isolation of therapeutic compounds (Newman & Cragg, 2012). As a result, researchers are trying to look for new organisms which have the potential to produce novel antibiotics from unexplored habitats (Oskay, Tamer & Azeri, 2004). Aquatic microorganisms are of special interest as they have not been exploited extensively compared to terrestrial microbes (Zhang et al., 2005).

Many molecular techniques have been developed in recent years for assessing genomic diversity of bacteria. Molecular identification of bacteria was performed by 16S rRNA gene sequencing as it is most conserved region and less prone to mutations (Kaushlesh et al., 2012). PCR fingerprinting methods like enterobacterial repetitive intergenic consensus (ERIC)-PCR and BOX-PCR has been extensively used to study genetic relationship as they have discriminatory capability in differentiating different genera of bacteria (Versalovic, Koeuth & Lupski, 1991; Rademaker et al., 2000). Tamdil is a reservoir freshwater lake situated 110 kms from Aizawl, the capital of Mizoram, North East India. The lake is reconstructed as a part of building fishing reservoirs by the Fisheries Department, Government of Mizoram, and is one of the 115 wetlands in India identified under the National Wetland Conservation Programme (vide D.O.No.J/2201/01/10-CS(W)).

The aim of the present study was to assess the diversity of cultivable bacteria from the sediments of the freshwater Tamdil Lake, to study their antimicrobial activities, their resistant to frequently used antibiotics and genetic relationship among the organisms. Investigating the bacterial population in fresh water sediments is of great importance in the general understanding of the aquatic ecosystem.

Materials and Methods

Sample collection

Water sediment samples were collected from five different locations of Tamdil Lake (23°44′20.4″N and 92°57′10.8′E) during the month of April and May 2013 in a sterile screw capped tube and brought into Molecular Microbiology and Systematics Laboratory, Department of Biotechnology, Mizoram University (Fig. S1). The samples were stored at 4°C till processed.

Isolation and antibiotic susceptibility profiling

Isolation was done using serial dilution method as described by Brown (2005) with few modifications. Purified bacterial isolates were grown in nutrient broth and incubated at 37°C, 150 rpm for 72 h. The grown bacterial suspension was spread on Muller Hinton agar (MHA) media using a sterile spreader. The plates were dried before placing the diffusion discs containing antibiotics. Susceptibility of the isolates to 12 different antibiotics was performed following Kirby Bauer disk diffusion method as described by Robert et al. (2009) as per National Committee for Clinical Laboratory Standards (NCCLS). Commercially available discs containing gentamicin (10 µg); Norfloxcin (30 µg); tetracycline (30 µg); ampicillin (10 µg); erythromycin (15 µg); streptomycin (30 µg); methicillin (5 µg); ofloxacin (5 µg); kanamycin (5 µg); furazolidone (50 µg); ketoconazole (50 µg) and nitrofurantoin (200 µg) were used. Antibiotic sensitivity was observed by measuring the diameters of the inhibition zone in mm and categorized as resistant, intermediate and sensitive to antibiotics.

Genomic DNA extraction, 16S rRNA gene amplification and phylogenetic analysis

Genomic DNA was isolated by using bacterial DNA extraction kit (Invitrogen Life technologies) as per manufacturer’s protocol. The 16S rRNA gene fragment was amplified by using universal primers—PA: 5′-AGA GTT TGA TCC TGG CTC AG-3′) and PH: 5′-AAG GAG GTG ATC CAG CCG CA-3′ (Qin et al., 2009). Reaction was carried out in a total volume of 25 µl consisting 1.0 µl of genomic DNA (50 ng), 0.2 µl of each primer (10 pmol), 2.0 µl of deoxynucleotide triphosphates (2.5 mM each), 2.5 µl of 1X PCR buffer, 0.2 µl of Taq DNA polymerase (1 U/ µl) and 15.9 µl MilliQ grade water. PCR was performed on Veriti thermal cycler (Applied Biosystem, Singapore) under following conditions: initial denaturation at 95°C for 4 min, followed by 30 cycles of denaturation at 94°C for 30 s, annealing at 57.5°C for 40 s and extension at 72°C for 1.3 min with a final extension step at 72°C for 10 min. A negative control reaction mixture without DNA template of bacteria was also included with each set of PCR reactions. The amplified PCR product was run on 1.5% agarose gel and visualized under gel documentation system Bio-Rad XR+ system (Hercules, CA, USA). The amplified products were purified using Purelink PCR Purification Kit (Invitrogen Life technologies) and were sequenced commercially at Sci-Genome Labs Pvt. Ltd, India.

ERIC-PCR fingerprinting

The primer sequences ERIC-1R (5′-CACTTAGGGGTCCTCGAATGTA-3′) and ERIC-2F (5′-AAGTAAGTGACTGGGGTGAGCG-3′) were used to amplify the regions of bacterial genome positioned between the ERIC sequences as described by Versalovic, Koeuth & Lupski (1991). PCR amplification was carried out on a Veriti thermal cycler (Applied Biosystem, Singapore) in a total reaction volume of 25 µl. The reaction mixture consist of DNA template (50 ng)—2.5 µl, 10X reaction buffer—2.5 µl, dNTP mix (10 mM)—2 µl, 10 pmol of each primer (ERIC 1R and ERIC 2F), 1 µl of MgCl2 (25 mM), and 2 U of Taq DNA polymerase (In-vitrogen, USA). PCR was performed under following conditions: initial denaturation at 95°C for 7 min and then subjected to 30 cycles of denaturation at 94°C for 1 min, annealing at 50°C for 1 min and extension at 65°C for 8 min with a final extension step at 65°C for 16 min. A negative control reaction mixture without DNA template of bacteria was also included with each set of PCR reactions. The amplified products were separated by electrophoresis on a 2% agarose gel using 1X TAE buffer. The PCR bands were analyzed under UV light and documented using a BioRad Gel Doc XR+ system (Hercules, CA, USA).

BOX-PCR fingerprinting

BOXA1R PCR fingerprinting was done using primer sequences BOXA1R (5′-CTACGG CAAGGCGACGCTGACG-3′ as described by Rademaker et al. (2000). PCR amplification was performed in a total volume of 25 µl reaction mixture, containing 50 ng of genomic DNA, 2.5 µl of 10X Taq Buffer, 1.5 µl of MgCl2, 2.0 µl of 2.5 mM dNTPs (2.5 mM), 1 µl of 10 pmol BOXA1R primer, 1 µl of DMSO (10%), 0.5 µl of BSA (10 mg/ml) and 1 µl of 2 U Taq DNA polymerase. The DNA was amplified under the following conditions: initial denaturation at 95°C for 7 min followed by 30 cycles at 94°C for 1 min, at 55°C for 1 min, and at 65°C for 8 min with a final extension step at 65°C for 10 min. The amplified products were separated and visualized as stated above.

Screening for antibacterial activity

The selected isolates were screened for their antibacterial activity against three pathogenic bacterial strains; Staphylococcus aureus (MTCC-96), Pseudomonas aeruginosa (MTCC-2453) and Escherichia coli (MTCC-739), all were obtained from the Microbial Type Culture Collection, Institute of Microbial Technology (IMTECH), Chandigarh. Pure isolates were grown in nutrient broth for extract preparation. The grown cultures were centrifugation at 8,000 rpm for three min and the supernatant was used for screening of antimicrobial activity by using agar well diffusion method (Saadoun & Muhana, 2008). The test pathogenic bacteria were spread on nutrient agar plate and wells were prepared using sterile cork borer of 6 mm diameter. A total of 50 µl clear supernatant of bacterial isolates were dispensed into each wells and the plates were incubated at 37°C for 24 h. The antimicrobial activities of the isolates were observed by measuring the diameter of the inhibition zone around each well.

Screening for antifungal activity

All the isolates were screened for their antagonistic activity against three plant pathogenic fungi viz. Fusarium proliferatum (MTCC-286), Fusarium oxysporum (CABI-293942) and F.usarium oxy. ciceri (MTCC-2791) by dual culture in vitro assay (Bredholdt et al., 2007). All plates were inoculated at 28°C for seven days and percentage of inhibition was calculated by using the formula: C − T∕C ×100, where, C is the colony growth of fungal pathogen in control, and T is the colony growth in dual culture.

Detection of biosynthetic gene sequences (PKS II and NRPS)

The potential antagonistic isolates were subjected for the amplification of genes for KS domains of Polyketide synthase (PKS-II) and the adenylation domains of non-ribosomal peptide synthetase (NRPS). NRPS gene fragments were amplified using degenerate primers: A3F 5′-GCSTACSYSATSTACACSTCSGG-3′ and A7R 5′-SASGTCV CCSGTSGCGTAS-3′ (Ayuso-Sacido & Genilloud, 2005). The degenerate primers, KS1F 5′-TSGCSTGCTTGGAYGCSATC-3′ and KS1R 5′-TGGAANCCGCCGAABCCTCT-3′, were used for amplifying PKS-II (Yuan et al., 2014). The PCR products were visualized under gel documentation system as stated above.

PKS II (50 μL)

For PKS II gene amplification, we used 3 µL of template DNA, 5 µL 10X buffer, 1 µL of MgCl2 (25 mM), 1 µL DMSO (10%), 5 µL of dNTP (2.5 mM), 1.8 µL each primer (10 mM) and Taq DNA polymerase (2U). PCR conditions as follows: 5 min at 95°C, followed by 35 cycles of 1 min at 95°C, 1 min 30 s at 58°C and 2 min at 72°C, followed by a 10-min extension at 72°C.

NRPS (50 μL)

For NRPS gene amplification, we used 3 µL template DNA, 5 µL 10X buffer, 1 µL of MgCl2 (25 mM), 1 µL DMSO (10%), 5 µL dNTP (2.5 mM) 2 µL each primer (10 mM) and Taq DNA polymerase (2U). PCR conditions as follows: 5 min at 95°C, followed by 35 cycles of 1 min at 95 °C, 2 min at 59°C and 4 min at 72°C, followed by 10 min extension at 72°C.

Statistical Analysis

DNA Sequences were compared with NCBI GenBank database using BlastN search program and sequences were aligned using the Clustal W software packaged in MEGA 5.05 (Thompson et al., 1997; Kumar et al., 2012). Suitable model was selected according to the lowest BIC (“Bayesian Information Criterion”) and the highest AIC (“Akaike Information Criterion”) scores. The phylogenetic tree was constructed by Neighbour joining method using MEGA 5.05 with Kimura 2-parameter model (R = 1.53) (Kimura, 1980), taking E. coli as an out group. The robustness of the phylogenetic tree was tested by bootstrap analysis using 1,000 replicates using p-distance model (Felsenstein, 1985).

Polymorphic DNA band were recorded in binary form i.e., 1 in case of presence of band and 0 when there is no band, to generate a binary matrix (Sneath & Sokal, 1973) for ERIC and BOX. The binary matrix was used to calculate the Simple Matching (SM) coefficient, phylogenetic tree was constructed using the Unweighted Pair Group Method with Arithmetic Mean (UPGMA) method (Lopez & Alippi, 2009) using Numerical Taxonomy SYStem (NTSYS version 2.2).

Results

Isolation of bacteria

A total of fifty cultivable bacteria were isolated from the sediment of Tamdil Lake, exhibiting distinct colony characteristics like size, opacity, pigmentation and texture. The colonies of bacterial strains were soft, sticky and also observed with white, red and yellow colours. The microscopic analysis indicates that 67% of the isolates were gram positive and 33% were gram negative bacteria.

Antibiotic sensitivity assay

A total of 12 standard known antibiotics were used to screen bacterial strains for their antibiotic sensitivity pattern (Table 1). Out of 50 bacteria examined, 33 strains showed resistance to at least two of the antibiotics tested. All the selected isolates showed resistance against methicillin and ampicillin (100% each). Most of the isolates were susceptible to tetracycline (n = 32) followed by norfloxacin (n = 29), genatamycin (n = 27), furazolidone (n = 24) and ketoconazole (n = 23). The parentage degree of resistance to erythromycin, streptomycin, kanamycin, nitrofurantoin and ofloxacin was 51.5, 48.4, 39.3, 33.3 and 24.2, respectively. All the isolates were more sensitive against gentamicin except BPSWAC14, 83 and 109. The isolates BPSWAC9, 14, 82, 83, 84, 108 and 109 showed resistance against six out of 12 antibiotics tested which might be good candidates for antibiotics production.

Table 1 Antibiotic sensitivity profile of bacterial isolates against 12 tested standard antibiotics.

Isolate No	Antibiotic susceptibility	
	A	TE	Gen	M	Of	Nx	E	S	K	Fr	Kt	Nf	
BPSWAC1	R	S	S	R	S	S	S	S	S	S	S	S	
BPSWAC3	R	S	I	R	I	I	S	I	I	S	S	I	
BPSWAC4	R	S	S	R	S	S	I	R	S	I	S	S	
BPSWAC5	R	S	S	R	R	S	R	I	S	R	S	S	
BPSWAC6	R	S	S	R	S	S	S	R	S	S	I	S	
BPSWAC7	R	S	S	R	R	S	S	S	S	S	S	R	
BPSWAC9	R	S	S	R	S	S	R	S	R	R	R	S	
BPSWAC14	R	S	R	R	S	S	R	R	R	S	S	S	
BPSWAC20	R	I	S	R	S	I	I	S	R	S	R	S	
BPSWAC26	R	S	S	R	R	S	I	R	R	R	S	R	
BPSWAC39	R	S	S	R	S	S	S	R	R	R	S	S	
BPSWAC41	R	S	S	R	R	S	R	S	S	S	R	I	
BPSWAC79	R	S	S	R	S	S	R	I	S	R	S	S	
BPSWAC80	R	S	S	R	S	S	S	S	S	I	S	S	
BPSWAC81	R	S	I	R	S	S	S	S	S	S	S	R	
BPSWAC82	R	S	S	R	S	R	R	S	R	R	S	S	
BPSWAC83	R	S	R	R	S	S	S	R	R	S	R	S	
BPSWAC84	R	S	S	R	R	S	R	R	S	S	S	R	
BPSWAC85	R	S	S	R	S	S	R	S	S	S	S	S	
BPSWAC90	R	S	S	R	S	S	S	R	R	S	R	R	
BPSWAC91	R	S	S	R	S	S	R	I	S	I	S	S	
BPSWAC92	R	S	S	R	S	S	S	R	I	S	S	S	
BPSWAC93	R	S	S	R	S	S	R	S	S	S	S	R	
BPSWAC94	R	S	S	R	S	S	S	R	S	S	I	R	
BPSWAC107	R	S	S	R	S	S	S	S	R	S	S	S	
BPSWAC108	R	S	S	R	S	R	R	S	S	S	R	R	
BPSWAC109	R	S	R	R	S	S	R	R	R	S	S	S	
BPSWAC110	R	S	S	R	R	S	S	S	S	S	S	S	
BPSWAC111	R	S	S	R	S	S	I	S	S	S	R	R	
BPSWAC112	R	S	R	R	S	S	S	I	R	S	S	S	
BPSWAC113	R	S	S	R	S	S	S	S	S	S	S	S	
BPSWAC114	R	S	S	R	S	S	S	S	S	S	S	S	
BPSWAC115	R	S	S	R	I	S	I	S	S	S	R	S	
Notes.

TITLE Degree of sensitivity: >10 mm Sensitive

5.0–9.9 mm intermediate

0.0–4.9 mm resistant

TITLE Gen Gentamicin (10 μg)

Nx Norfloxcin (30 μg)

T Tetracycline (30 μg)

A Ampicillin (10 μg)

E Erythromycin (15 μg)

S Streptomycin (30 μg)

M methicillin (5 μg)

Of ofloxacin (5 μg)

K Kanamycin (5 μ)

Fr Furazolidone (50 μg)

Kt Ketoconazole (50 μg)

Nf Nitrofurantoin (200 μg)

ERIC-PCR fingerprinting

The ERIC-PCR fingerprinting of the isolates yielded a discriminatory patterns with genomic size ranging from approx. <100 bp to 3.0 kb. A dendrogram (Fig. 1A) was constructed by using Jaccard similarity coefficients and the UPGMA method. Dendrogram generated by ERIC-PCR divided the isolates into two clusters (A & B). Cluster A was bigger and divided into 2 sub-clusters (A1 and A2). Cluster A consist of 25 isolates belonging to the genus Staphylococcus, Lysinibacillus, Azospirillum, Bacillus, Bacterium and Aneurinibacillus. Cluster B consist of 8 isolates comprising different genera belonging to Novosphingobium, Bosea, Methylobacterium and Achromobacter. Isolates BPSWAC82 and BPSWAC93 showed 100% similarity and were identified as Methylobacterium based on 16S rRNA gene sequence. This result agreed with the phylogenetic tree of the 16S rDNA with bootstrap supported value of 88%.

Figure 1 Dendrogram generated from (A) ERIC-PCR and (B) BOX-PCR genomic fingerprints of bacterial isolates using Ntsys 2.0.

BOX-PCR fingerprinting

BOX-PCR fingerprinting of a total of 33 isolates showed specific patterns corresponding to particular genotypes and size recognizable bands were between <100 bp to 3 kb. A dendrogram generated from BOX-PCR analysis comprised of two major clusters (A & B). Cluster A was larger containing 24 isolates whereas cluster B consist of nine isolates. Cluster A is divided into 2 sub-cluster (A1 and A2). Staphylococcus and Lysinibacillus were grouped together in cluster A1 whereas Achromobacter, Bacillus and Aneurinibacillus were grouped together in cluster A2. Cluster B is also divided into two sub-cluster (B1 and B2) comprising of different genera belonging to Bacterium, Azospirillum, Methylobacterium, Bosea and Novosphingobium (Fig. 1B). BOX-PCR analysis demonstrated high discriminatory ability by constructing genus-specific clusters that were able to differentiate all the different genera reported in this study.

Evaluation of antimicrobial activity

The selected 33 isolates based on their antibiotic susceptibility profile were tested for antimicrobial activities against bacterial pathogens (Staphylococcus aureus, Pseudomonas aeruginosa and Escherichia coli) and fungal pathogens (Fusarium proliferatum, F. oxysporum and F. oxy. ciceri). Out of 33 isolates, nine strains exhibited antibacterial activity against two tested pathogens (Table 2). Only two isolates BPSWAC82 and BPSWAC108 showed positive activity against all the bacterial and fungal pathogens. Isolate BPSWAC82 showed highest activity against F. oxysporum (73.68%) and F. proliferatum (73.68%), whereas BPSWAC14 was found high activity against F. oxy.ciceri (64.38%). On the other hand, isolate BPSWAC20 had acute activities against bacterial pathogens E. coli (10 mm), whereas BPSWAC109 exhibited highest activity against S. aureus (11 mm) (Table 2).

Table 2 In vitro antagonistic activity of selected bacterial isolates against fungal and bacterial pathogens and detection of biosynthetic genes.

Isolate no.	NCBI accession no	Percentage of inhibition (PI ± SD)	Zone of inhibition in mm (ZI ± SD)	PKS II	NRPS	
		F. oxysporum	F. oxy. ciceri	F. proliferatum	E. coli	P. aeruginosa	S. aureus			
BPSWAC9	KM243385	0.00a	0.00a	0.00a	9 ± 0.1a	6 ± 0.29a	9 ± 0.18a	−	−	
BPSWAC14	KM405299	60.50 ± 0.00bc	64.38 ± 0.17bc	0.00a	9 ± 0.1a	12 ± 0.2bc	10 ± 0.05bc	+	+	
BPSWAC20	KM405305	0.00a	58.9 ± 0.26bde	0.00a	10 ± 0.1bc	8 ± 0.10bde	9 ± 0.2a	−	+	
BPSWAC82	KR857324	73.68 ± 0.012bde	61.64 ± 0.02bdfg	73.68 ± 0.12bc	9 ± 0.05a	12 ± 0.15bc	9 ± 0.05a	+	+	
BPSWAC83	KR857325	65.78 ± 0.28bdfg	0.00a	0.00a	7 ± 0.25bde	8 ± 0.05bde	−	−	−	
BPSWAC84	KR857326	0.00a	39.72 ± 0.001bdfhi	47.36 ± 0.02bde	9 ± 0.0a	12 ± 0.3bc	10 ± 0.2bc	+	+	
BPSWAC90	KT232317	47.36 ± 0.03bdfhi	45.2 ± 0.02bdfhj	47.36 ± 0.14bde	8 ± 0.02bdf	7 ± 0.15bdef	−	+	+	
BPSWAC108	KT429618	60.50 ± 0.03bc	39.72 ± 0.09bdfhi	47.36 ± 0.08bde	9 ± 0.02a	6 ± 0.15a	9 ± 0.1a	+	+	
BPSWAC109	KT429619	50 ± 0.045bdfhj	0.00a	0.00a	9 ± 0.1a	9 ± 0.05bdeg	11 ± 0.1bd	+	+	
Notes.

Mean (±SD) followed by the same letter(s) in each column are not significantly different at P < 0.05 using Duncan’s new multiple range test, (+) and (−) indicates the presence and absence of PKS II and NRPS genes.

Detection of PKS and NRPS genes in selected strains

Six isolates out of 33 strains visualised band in type II polyketide synthases (PKS-II) with an amplification size of 600 bp (Fig. S2), whereas nonribosomal peptide synthetases (NRPS) genes were detected in seven isolates (21.2%) with expected size of 700 bp (Fig. S2). Isolates BPSWAC14, 82, 84, 90, 108 and 109 showed positive amplification products with both the degenerate primers for PKSII and NRPS respectively (Table 2). Isolates BPSWAC21 showing band against NRPS primers also indicated the highest antimicrobial activities against E. coli pathogens. Isolate BPSWAC14 identified as Novosphingobium sp. and showed antimicrobial biosynthetic potential in both genes. This isolates was found as rare genera among them and may be useful for isolation of natural products.

Sequence alignment and phylogenetic analysis

The isolated 33 strains were sequenced by amplification of 16S rRNA gene. All the partial 16S rRNA sequences were aligned using BLAST analysis and deposited in NCBI GeneBank having an accession no. The results showed that the isolates were classified into ten different genera. Majority of the isolates belongs to Staphylococcus (25%), followed by Bacillus (21%), Lysinibacillus (12%), Achromobacter (9%), Bacterium (9%) Methylobacterium (6%), Bosea (6%), Aneurinibacillus (6%), Azospirillum (3%) and Novosphingobium (3%) (Fig. 2). The 16S rRNA gene sequences by BlastN exhibited high level of sequence 98–100% similarity confirmed that seven isolates could be members of genus Staphylococcus. The sequences of the three isolates (BPSWAC5, BPSWAC6 and BPSWAC9) showed high identity (99%) to the genus Bacterium whereas isolate BPSWAC14 and BPSWAC26 exhibited high identity (88%) to the genus Novosphingobium and Achromobacter respectively. Maximum-likelihood and neighbor-joining methods were used for the construction of phylogenetic tree. The phylogetic tree generated by both methods showed that all Staphylococcus strains as well as other genera except Novosphingobium sp. forms a major clade I along with the type strains retrieved from databases. Most of the rare genera like Novosphingobium sp., Bosea sp. and Methylobacterium sp. clustering to form another clade II in Maximum likelihood tree (Fig. 3B) under the bootstrap value of 30% respectively. However, the neighbour joining tree did not cluster Bosea sp. and Methylobacterium sp. together in clade II (Fig. 3A).

Figure 2 Pie chart showing the distribution of bacteria in water sediment of Tamdil Lake.

Figure 3 (A) Neighbor-joining phylogenetic tree based on 16S rRNA gene of bacteria identified from Tamdil Lake. (B) Maximum likelihood phylogenetic tree based on 16S rRNA genes. Numbers at branches indicate bootstrap values of neighbour-joining analysis (>50%) from 1,000 replicates.

Nucleotide sequence accession numbers

All the isolates were identified by sequencing 16S rRNA gene and the sequences were deposited in NCBI-GenBank and the accession numbers of bacterial isolates are KM243378; KM243379; KM243380; KM243381; KM243382; KM243383; KM243385; KM405299; KM405305; KM405311; KR703476; KR703477; KR857321; KR857322; KR857323; KR857324; KR857325; KR857326; KR857327; KT232317; KT232318; KT232319; KT232320; KT232321; KT429617; KT429618; KT429619; KT429620; KT429621; KT429622; KT429623; KT429624; KT429625.

Discussion

There is an urgent need of new and novel antimicrobials with the development of multiple drug resistant microbes (Wise, 2008). Several studies were carried out by various researchers to investigate the occurrence and distribution of antibiotics resistant bacteria in water ecosystems (Baya et al., 1986; Herwig, Gray & Weston, 1997; Mudryk & Skorczewski, 1998). The chances of finding new bioactive compounds is much higher from the bacteria isolated from unexplored habitats (Bredholdt et al., 2007) which have become significant for discovering novel compounds (Saadoun & Gharaibeh, 2003).

In this study, an attempt has made to isolate bacteria from the sediment of Tamdil Lake, a wetland present in Mizoram, Northeast India. Water sediments contain around 30% of the earth’s biomass and are a key ecological niche for novel microorganisms (Whitman, Coleman & Wiebe, 1998). We reported 50 cultivable bacteria isolated from a wetland located in Mizoram, Northeast India. Microscopic analysis confirmed 67% as gram positive and 33% as gram negative bacteria which is in agreement with the finding of Zhuang et al. (2003) and Gontang, Fenical & Jensen (2007).

All the isolates were determined for their antibiotics susceptibility profiling using twelve standard antibiotic impregnated discs. We detected significant antibiotic resistance to most of the antibiotics under investigation. Thirty-three isolates were selected which showed resistance to methicillin and ampicillin (100% each). All the isolates were resistant to ampicillin supported by the findings of Falcao et al. (2004), Scoaris et al. (2008) and Reboucas et al. (2011) and were susceptible to tetracycline (96.6%). Among 33 isolates, Bacterium (BPSWAC9), Novosphingobium (BPSWAC14), Methylobacterium organophilium (BPSWAC82), Lysinibacillus sp. (BPSWAC83), Bosea sp. (BPSWAC84), Aneurinibacillus aneurinilyticus (BPSWAC108) and Bacillus sonorensis (BPSWAC109) showed resistant against 6 out of 12 antibiotics tested and might be a good candidate for further investigation. To best of our knowledge, this is the first time reported that Novosphingobium sp. (BPSWAC14), Methylobacterium organophilium (BPSWAC82), and Bosea sp. (BPSWAC84) showed multi antibiotics resistance. Multiple drug resistance of these isolates could be due to some pollutants in the lakes. Multiple bacterial resistances to antibiotics had earlier been reported in aquaculture environments (Hatha et al., 2005).

The genomic relatedness of the selected isolates was studied using ERIC and BOX-PCR fingerprinting. The dendrogram generated by ERIC-PCR divided the isolates into two groups (A and B), Staphylococcus, Lysinibacillus, Azospirillum, Bacillus, Bacterium and Aneurinibacillus falls under cluster A. Cluster B consist of Novosphingobium, Bosea, Methylobacterium and Achromobacter, which was in agreement with the findings of De-Bruijn (1992). ERIC-PCR has previously demonstrated useful for genotyping Vibrio parahaemolyticus as well, isolated from aquatic system in North China (Xu et al., 2016). The dendrogram generated by BOX-PCR also consist of cluster A and B. Staphylococcus, Lysinibacillus, Achromobacter, Bacillus and Aneurinibacillus were grouped together in cluster A. Cluster B of different genera belonging to Bacterium, Azospirillum, Methylobacterium, Bosea and Novosphingobium which was in agreement with the findings of Lee et al. (2012). BOX-PCR fingerprinting shows that it is very useful technique to differentiate between very closely related bacterial strains identification and has been applied to study the great genetic diversity at species level (Versalovic, Koeuth & Lupski, 1991). In this study, we observed that genetic variation was very high among the 33 isolates, when analyzed by BOX-PCR fingerprinting and among them Staphylococcus sp. was the dominant species (25%). The finding from our study was in agreement with previously reports of Ali (2014).

A significant antimicrobial activity of bacteria was detected in this study against both Gram positive and Gram negative bacteria. Interestingly, those seven isolates that showed most resistance against antibiotics also showed antagonistic activity also. Two isolate Methylobacterium organophilum (BPSWAC82) and Aneurinibacillus aneurinilyticus (BPSWAC108) showed positive activity against all the bacterial and fungal pathogens. BPSWAC82 showed highest activity against F. oxysporum (73.68%) and F. proliferatum (73.68%). In addition, the anti fungal activity of Methylobacterium spp. against Fusarium udum, F. oxysporum, Pythium aphanidermatum, and Sclerotium rolfsii was also reported (Poorniammal, Sundaram & Kumutha, 2009). Bacillus sonorensis (BPSWAC109) exhibited highest activity against S. aureus which was in accordance with (Rakesh et al., 2011). A significant antibacterial activity of the genus Bacillus was also reported recently (Etyemez & Balcazar, 2016) and several species of Bacillus produce antimicrobial peptides which are commercially available (Leaes et al., 2016).

Nonribosomal peptides and polyketides are two different families of natural products which are the major source of pharmaceutical products (Walsh, 2004), synthesised by nonribosomal peptide synthetases (NRPS) and polyketide synthases (PKS) respectively (Khater, Anand & Mohanty, 2016). Detection of these genes has been generally used for assessing biosynthetic potential of culturable and non-culturable microorganisms (Minowa, Araki & Kanehisa, 2007). In this study, PKS type II and NRPS genes were detected in six isolates and seven isolates respectively. Interestingly, biosynthetic genes were detected in those isolates showing antimicrobial activities which further proved the existence of biosynthetic gene clusters and may be responsible for the production of antimicrobial secondary metabolites. Both PKS II and NRPS genes were detected in the isolated strains like Bacillus sonorensis (BPSWAC 109), Aneurinibacillus aneurinilyticus (BPSWAC 108), lysinibacillus fusiformis (BPSWAC 90), Bosea sp. (BPSWAC 84), Methylobacterium organophilium (BPSWAC 82) and Novosphingobium sp (BPSWAC 14). Earlier studies reported the presence of biosynthetic genes like PKS and NRPS in Bacillus spp. and some other firmicutes (Straight et al., 2007; Aleti, Sessitsch & Brader, 2015) but, to our knowledge this is the first time for the report of PKS II and NRPS gene in Novosphingobium and Bosea sp.

All the bacterial isolates were characterized by sequencing the 16S rRNA gene; most isolates showed 98–100% identity with NCBI BlastN sequences. Bacteria (n = 33) isolated from the sediment samples were diverse and represented two bacterial phyla (Proteobacteria and Firmicutes). Microorganisms belonging to Firimicutes (gram positive) were the leading group in the samples which was in accordance with the studies of Zhuang et al. (2003) and Gontang, Fenical & Jensen (2007) of marine sediments but in contrast with the findings of gram negative bacteria as a dominant bacteria isolated from lake water (Panneerselvam & Arumugam, 2012). Maximum-likelihood and neighbor-joining methods showed that all Staphylococcus strains as well as other genera except Novosphingobium sp. forms a major clade I with an exception of Bacterium morphologically similar with Staphylococcus sp. and hence clustered together along with the type strains. Most of the rare genera like Novosphingobium sp., Bosea sp. and Methylobacterium sp. clustering to form another clade II, consistent with the findings of previous studies (Shukla et al., 2011; Ganesan, 2013).

Conclusion

Fifty bacterial strains were isolated from the sediment of a freshwater lake and screened for antibiotic resistance and thirty-three isolates showed resistant against at least two of the selected twelve antibiotics. Genotyping using the 16S rRNA gene sequencing distinguished them into ten different genera with staphylococcus as dominant genus. Screening for anti-microbial activity revealed eight isolates with the potential to produce antimicrobials, which further proved the potential for the production of these compounds was further verified by the detection of PKS and NRPS biosynthetic genes. This is the first reported occurrence of two rare isolates (Novosphingobium and Bosea sp.) from a wetland with biosynthetic potential, which can be exploited for the search of biologically active metabolites

Supplemental Information

Figure S1 Map showing the location of the sampling site

Click here for additional data file.

Figure S2 Detection of biosynthetic genes (A) PKS-II and (B) NRPS using degenerate primers

Click here for additional data file.

The authors are thankful to Dr. Anthonia O’Donovan, School of Natural Sciences, National University of Ireland, Galway for the critical reading and the language editing of the manuscript. Authors are thankful to North Eastern Hill University, Shillong, for SEM.

Additional Information and Declarations

Competing Interests

Author Contributions

DNA Deposition

Vijai K. Gupta is an Academic Editor for PeerJ.

Zothanpuia and Ajit K. Passari performed the experiments, contributed reagents/materials/analysis tools, wrote the paper, prepared figures and/or tables.

Vijai K. Gupta analyzed the data, reviewed drafts of the paper.

Bhim P. Singh conceived and designed the experiments, analyzed the data, contributed reagents/materials/analysis tools, wrote the paper, reviewed drafts of the paper, editing of the manuscript.

The following information was supplied regarding the deposition of DNA sequences:

All the isolates were identified by sequencing the 16S rRNA gene. The sequences were deposited in NCBI-GenBank: KM243378; KM243379; KM243380; KM243381; KM243382; KM243383; KM243385; KM405299; KM405305; KM405311; KR703476; KR703477; KR857321; KR857322; KR857323; KR857324; KR857325; KR857326; KR857327; KT232317; KT232318; KT232319; KT232320; KT232321; KT429617; KT429618; KT429619; KT429620; KT429621; KT429622; KT429623; KT429624; KT429625.

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
