# Peer review of "Detection of antibiotic-resistant bacteria endowed with antimicrobial activity from a freshwater lake and their phylogenetic affiliation"

_PeerJ, doi:10.7717/peerj.2103_

## Round 0.1 · original submission · Minor Revisions

· Academic Editor

Minor Revisions

Please, pay attention to the comments of reviewer #1 and come with a version that takes them into account. If you disagree, tell us why. So, please, submit a detailed rebuttal. Your new version may be subjected to further review, which means that I cannot give any guarantee that the paper will eventually be accepted.

Reviewer 1 ·

Basic reporting

Manuscript is well written and findings are good. Authors used advance methods to establish facts. Please add few references of 2016.

Experimental design

Experimental design should be more appropriate and elaborated. Please add more details in the experimental section.

Validity of the findings

Findings are good and well established.

Additional comments

No comments.

Reviewer 2 ·

Basic reporting

The manuscript “Detection of antibiotic resistant bacteria endowed with antimicrobial activity from fresh water lake and their phylogenetic affiliation" relates to an interesting topic of microbiology. Generally speaking the study looks sound and the data could be useful for the development of novel antibiotics against resistant organisms .The study is descriptive and has characterize the antibiotic resistant bacteria, which is a need for the day for the development of antibiotics. The authors have characterized the isolates based on their morphology and 16S rRNA gene sequence comparison with the database. Isolates were clustered using ERIC and BOX PCR profiles. Isolates were screened for antimicrobial activity as well as for the presence of two biosynthetic genes (PSKII and NRPS). The results are interesting and novelties of this work are the isolation of antibiotic resistant bacteria from the wet land of Mizoram, India, which were further characterize and shown the ability of antibiotic resistant and were characterize molecularly, which is quite appreciated. Authors also have been characterizing first time two isolates i.e. Novosphingobium and Bosea sp. from a wetland with biosynthetic potential, which can be exploited for the search of biologically active metabolites. Present work carried out by Zothanpuia et al. Is interesting and may be considered for publication in PeerJ.

Experimental design

The experimental design is appropriate and authors have attempted required methodology for the proposed work. Material and methods is well presented and addressed all standard references for methodology.

Validity of the findings

Not required

Additional comments

Head wise description is stated below

Abstract – Abstract is well design and well presented with most of the findings as stated in the manuscript. Also, highlight the aims of this study and the fact that these isolates can be used exploited for the search of biologically active metabolites. Abundance is also mentioned nicely which is an add-on to the work.

Introduction - Background information is nicely presented. Findings from previous studies are fully presented. I recommend elaborating a bit more on the literature here presented though I understand that LAM has world limit. The topic had introduced pretty well with the supported literature and I do also believe that bacterial population in fresh water sediments is of great importance in general understanding the aquatic ecosystem.
Results and discussion:
The result headings are well presented and appropriate references has been cited in the discussion part.

---

## Round 0.2 · accepted · Accept

· Academic Editor

Accept

Although limited, the study provides interesting insights in the topic of study and should trigger larger scale investigations in this area.